# Retinal macrophage-like cell activation and ganglion cell layer thinning are associated with disability and MRI lesion burden in multiple sclerosis

Francesco Pichi [1,2]*, Yanny Perez Jimenez[2], Matteo Belletti[3], Alia Alsuwaidi[4], Fatema Alawadhi[4], Gina Lee[5], Piergiorgio Neri[2], Victoria Mifsud[5], Anu Jacob[5], Beatrice Benedetti[5], Ester Carreño[6]

1 University of Toronto, Department of Ophthalmology and Vision Sciences, Toronto, Canada, 2 Eye Institute, Cleveland Clinic Abu Dhabi, Abu Dhabi, United Arab Emirates, 3 IRCSS Azienda Ospedaliero-Universitaria di Bologna, Bologna, Italy, 4 Radiology Department, Cleveland Clinic Abu Dhabi, Abu Dhabi, United Arab Emirates, 5 Neurological Institute, Cleveland Clinic Abu Dhabi, Abu Dhabi, United Arab Emirates, 6 Hospital Universitario la Paz, Madrid, Spain

* ilmiticopicchio@gmail.com

## Abstract

### Background

Multiple sclerosis is a chronic neuroinflammatory disease in which microglia and macrophage-like cells play a central role in disease progression. The retina, as a central nervous system extension accessible to non-invasive imaging, offers a unique window into these processes.

### Objective

The aim of this study was to determine whether retinal macrophage-like cell characteristics and ganglion cell layer thickness are associated with clinical disability and magnetic resonance imaging lesion burden in patients with multiple sclerosis.

### Methods

In this cross-sectional observational study, patients with multiple sclerosis and age-matched healthy controls underwent retinal imaging using spectral-domain optical coherence tomography and swept-source optical coherence tomography angiography. Macrophage-like cells at the vitreoretinal interface were quantified using an automated image-processing pipeline, while ganglion cell layer and retinal nerve fibre layer thicknesses were measured using standardized segmentation. Clinical disability was assessed using the Expanded Disability Status Scale, the 9-Hole Peg Test, and the 25-Foot Walk. Brain magnetic resonance imaging lesion counts were obtained from T1- and T2-weighted sequences.

**Data availability statement:** Data are available upon reasonable request from the Cleveland Clinic Abu Dhabi Research Office (research@clevelandclinicabudhabi.ae). Data access is subject to institutional review and ethical approval, as the dataset contains potentially identifiable clinical information from patients enrolled under ethics protocol #A-2022-042.

**Funding:** The author(s) received no specific funding for this work.

**Competing interests:** The authors have declared that no competing interests exist.

**Abbreviations:** MS, multiple sclerosis; MLC, macrophage-like cell; OCT, optical coherence tomography; OCTA, optical coherence tomography angiography; GCL, ganglion cell layer; RNFL, retinal nerve fibre layer; EDSS, Expanded Disability Status Scale; 9HPT, 9-Hole Peg Test; T25W, Timed 25-Foot Walk; MRI, magnetic resonance imaging; VRI, vitreoretinal interface; SS-OCTA, swept-source OCT angiography.

## Results

Overall retinal macrophage-like cell parameters did not differ between patients with multiple sclerosis and controls. However, patients with greater disability, defined as an Expanded Disability Status Scale score of five or higher, showed significantly increased macrophage-like cell count and density. Macrophage-like cell size demonstrated a moderate positive association with T2 lesion burden, while only weak associations were observed with T1 lesions. Ganglion cell layer thickness was significantly reduced in the inferior and nasal macular regions in patients with multiple sclerosis compared with controls, whereas retinal nerve fibre layer thickness was preserved. Disability scores showed moderate correlations with upper and lower limb functional tests.

## Conclusions

These findings indicate that retinal cellular and structural alterations reflect both inflammatory and neurodegenerative components of multiple sclerosis. Retinal macrophage-like cell metrics are associated with disease burden and disability, while ganglion cell layer thinning provides evidence of neuroaxonal loss. Retinal imaging may serve as a non-invasive complement to magnetic resonance imaging for assessing disease severity and progression in multiple sclerosis.

## Introduction

Multiple sclerosis (MS) is a chronic autoimmune neurodegenerative disease characterized by inflammation and demyelination in the central nervous system (CNS) [1]. It affects approximately 2.8 million people worldwide [2]. MS typically manifests in individuals aged 20–50, with women being affected four times more frequently than men [3]. People of Northern European descent and white individuals have a higher risk of developing the disease, with prevalence decreasing in populations closer to the equator[4]. In the Middle East, the prevalence of MS in countries such as Kuwait, Qatar, Bahrain, and the United Arab Emirates ranges from 55 to 85 per 100,000 individuals [4–6].

In MS, the immune system erroneously targets the myelin sheath surrounding nerve fibers, driven primarily by autoreactive T-cells recognizing myelin proteins. Genetic predispositions, notably variations in the HLA DRB1 gene, combined with environmental factors such as Epstein-Barr virus infection, smoking, and vitamin D deficiency, contribute significantly to disease initiation [7].

Microglia, the resident immune cells of the CNS, originate from embryonic progenitors [8] and differ from peripheral macrophages due to their specialized adaptation to the tightly regulated CNS environment [9]. These cells are critical in neuroinflammation, demyelination, and axonal injury, often becoming activated early in the MS pathology, even prior to T-cell infiltration, as demonstrated in human and experimental autoimmune encephalomyelitis models [10,11].

Upon activation, microglia transition from a surveillant to a pro-inflammatory state, characterized by morphological changes and upregulated surface markers such as CD68 and MHC-II. They produce pro-inflammatory cytokines (e.g., TNF-α, IL-1β, IL-6), chemokines (e.g., CCL2, CXCL10), reactive oxygen species (ROS), and matrix metalloproteinases (MMPs) [12]. These mediators amplify local inflammation, facilitate peripheral immune cell entry by degrading the blood-brain barrier (BBB), and exacerbate neuronal and oligodendrocyte [13].

Microglia also interact with astrocytes, releasing inflammatory mediators like IL-1β and TNF-α, thus perpetuating neurotoxicity [10]. Moreover, microglial activation contributes to synaptic pruning via complement-mediated pathways, further exacerbating neuronal injury and highlighting their central role in MS pathogenesis [11,14].

Widespread microglial activation occurs in both lesional and normal-appearing white and gray matter regions in MS [15], with activated microglia displaying a reactive "ameboid" morphology that can yield neuroprotective or neurotoxic outcomes [16,17]. Functioning as antigen-presenting cells (APCs), these activated microglia present myelin antigens to infiltrating T-cells, facilitating BBB disruption and escalating inflammatory cascades [18]. Subsequent cytokine release by T-cells intensifies myelin damage and axonal injury, key drivers of progressive neurodegeneration in MS [19].

Histological studies have identified distinct microglial subtypes associated with specific MS pathologies, including neural cell loss and B-cell infiltration, emphasizing their potential roles in driving clinically silent progression and disability [20].

Macrophage-like cells (MLCs), including retinal microglia and infiltrating macrophages, are ramified, mobile cells present at the vitreoretinal interface [21]. Numerous studies utilizing OCT have reported increased MLC number, density, and size in various ocular conditions. For example, Carreño et al. observed an increased number and size of MLCs in toxoplasmosis-related ischemic areas [22]. Similarly, Pichi et al. identified elevated MLC counts in active posterior uveitis, suggesting their potential as biomarkers of inflammation and disease activity [23,24].

Given the retina's developmental and anatomical continuity with the CNS, retinal imaging offers a unique, non-invasive window into neuroinflammatory and neurodegenerative processes relevant to MS. Correlating retinal MLC metrics with MRI lesion burden and clinical disability may help bridge the gap between cellular-level pathology and conventional imaging biomarkers. In this study, we aim to analyze the characteristics of retinal MLCs detected using swept-source optical coherence tomography angiography (SS-OCTA), and SS-en face OCT in the retinas of MS patients, correlate them with MS radiological and neurological parameters, and compare these findings with those from healthy subjects. Our objective is to quantify MS-induced changes in the number and size of macrophage-like cells at the vitreomacular interface, providing sensitive biomarkers that may enhance early detection and support more effective treatment strategies for MS.

## Methods

### Study design and participants

This cross-sectional study included 100 consecutive patients diagnosed with multiple sclerosis (MS) according to the 2017 McDonald criteria and 51 healthy controls who presented at the Department of Neurology, Cleveland Clinic Abu Dhabi (CCAD), between March 2024 and October 2024. The study received ethical approval from the CCAD Ethics Committee (#A-2022–042) and adhered to the principles of the Health Insurance Portability and Accountability Act of 1996 and the Declaration of Helsinki. All participants provided written informed consent.

Inclusion criteria for MS patients were: [1] diagnosis of MS per the 2017 McDonald criteria [25]; [2] age between 18 and 60 years; and [3] an MRI of the brain with or without contrast performed within the past 12 months.

Exclusion criteria were: [1] ophthalmologic or systemic disorders affecting the retina (e.g., diabetes, uveitis) or conditions compromising retinal scan quality (e.g., cataracts, corneal opacities); [2] previous history of optic neuritis; and [3] incomplete medical release during enrollment.

For healthy controls, inclusion criteria included: [1] age between 18 and 60 years; [2] normal MRI findings; [3] no history of uveitis or vitreoretinal pathology; and [4] no personal or family history of MS.

## Neurological Evaluation

MS patients underwent comprehensive neurological assessments, including demographic and clinical data collection and a neurological examination to determine the Expanded Disability Status Scale (EDSS) score. The MS Functional Composite (MSFC) component tests were also collected. These included the Timed 25 Foot Walk (T25W) and 9-hole Peg Test (9HPT). Trained neurologists (BB, AJ, VM) conducted these evaluations at the CCAD MS clinic.

## MRI evaluation

Two trained physicians evaluated the lesion burden in the brain MRI of MS patients (AA and FA). Hyperintense T2/FLAIR lesions and hypointense T1 lesions were counted.

## Ophthalmic Examination

All participants underwent a comprehensive ophthalmic examination, including measurement of Snellen's best-corrected visual acuity (BCVA), which was later converted into logMAR for statistical analysis; spectral domain OCT imaging (Spectralis HRA OCT; Heidelberg Engineering, Heidelberg, Germany); and Plex Elite 9000 Swept-source OCTA for advanced retinal analysis.

## OCT Imaging

Spectral-domain OCT images were acquired for both eyes using a Spectralis OCT device with Glaucoma Module Premium Edition software (Heidelberg Engineering, Germany). Images were automatically segmented using vendor software (Heidelberg Eye Explorer v2.5.1) and manually corrected for segmentation errors. Image quality and segmentation accuracy were evaluated using the OSCAR-IB criteria [26]. Retinal thickness measurements were derived from segmented layer outputs.

## Macrophage-Like Cell (MLC) Analysis

MLC visualization was performed using a Plex Elite 9000 SS-OCTA machine operating at 100 kHz. A trained ophthalmologist (YP) obtained six high-quality OCTA scans (6 × 6 mm regions) centered on the fovea for each eye.

Acceptance criteria for scan quality included clear focus, minimal artifacts, no significant saccades, proper centration, regular illumination, and a signal strength of seven or higher. OCTA scans were exported from the superficial capillary plexus (the inner limiting membrane) to the inner plexiform layer for processing.

The vitreoretinal interface (VRI) slab was adjusted to a 3-µm depth (−7 to −10 µm from the inner limiting membrane) to minimize hyperreflectivity from the retinal nerve fiber layer and facilitate MLC visualization.

Image processing included the following steps:

1. Registration and Averaging: The "Register Virtual Stack Slices" plugin in FIJI (ImageJ) was used for elastic registration of six OCTA images, followed by averaging using the "AvgNoiseRmvr" plugin to enhance the signal-to-noise ratio. The same transformation matrix was applied to en-face OCT slabs, which were averaged to increase the signal-to-noise ratio.

2. Background Removal and Thresholding: Fast Fourier transform (FFT) filtering and automatic brightness thresholding were applied to highlight MLCs. Particles were visually marked in red.

3. Particle Analysis: The "Analyze Particles" filter was used to measure MLC size. Particles smaller than 576 µm², larger than 4896 µm², or with a circularity < 0.5 were excluded. Morphology-based thresholds were applied to match MLC characteristics.

All measurements were automated using an ImageJ Macro, that provided size and count data for further analysis (Fig 1).

## Statistical analysis

Statistical analyses evaluated the relationships between MLC parameters, clinical metrics, and structural outcomes in the MS and control groups. Descriptive statistics, including means and standard deviations, were calculated for age, gender distribution, and group size. For comparisons between groups (e.g., MS vs. controls), independent t-tests were used to analyze continuous variables such as MLC parameters, ganglion cell layer (GCL) thickness, and retinal nerve fiber layer (RNFL) thickness.

For categorical comparisons, a one-way analysis of variance (ANOVA) was performed to assess differences in MLC parameters based on treatment types and disease onset in the MS group. Pearson correlation coefficients were calculated to explore relationships between EDSS scores and clinical measures such as T1 and T2 lesions, 9-hole Peg Test times, and 25-foot walk times. Correlation analyses were also conducted to evaluate associations between MLC parameters and structural metrics (GCL and RNFL) for both eyes (OD and OS) of MS patients.

Where appropriate, data were visualized using bar plots, box plots, and scatter plots with regression lines to highlight trends and differences. All statistical analyses were performed using Python (pandas, scipy, and seaborn libraries), with a significance threshold of $p < 0.05$. For all correlation analyses between MLC parameters and MRI/clinical variables (EDSS, T1 and T2 lesion counts), the unit of analysis was the individual patient, with MLC metrics averaged across both eyes for each subject. This averaging approach was justified by the moderate-to-strong inter-eye correlation observed for MLC count ($r = 0.635$, $p < 0.001$). For GCL and RNFL thickness comparisons between MS patients and controls, eye-level data were used. To account for multiple comparisons, Bonferroni correction was applied as follows: for the six MLC–MRI correlation analyses (3 MLC parameters × 2 MRI measures), the adjusted significance threshold was $p < 0.0083$; for the four regional GCL comparisons, the threshold was $p < 0.0125$; and for the six regional RNFL comparisons, the threshold was $p < 0.0083$.

## Results

We included 195 eyes from 100 patients in the MS group and 102 eyes from 51 patients in the control group. The proportion of females was 62% in the MS group and 76.5% in the control group. The mean age was 35.08 years (SD = 8.67) in the MS group and 35.71 years (SD = 9.54) in the control group. There was no significant difference in age (p = 0.69) between the two groups.

### MS group analysis

The MS group comprised primarily patients with relapsing-remitting phenotype, representing 91.0% of the cohort, followed by secondary progressive (4.0%) and primary progressive (2.0%) phenotypes, with minor entries for other

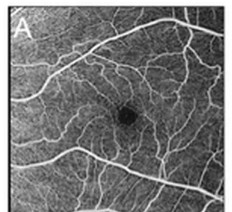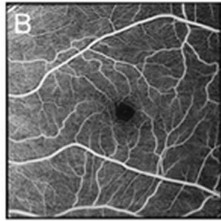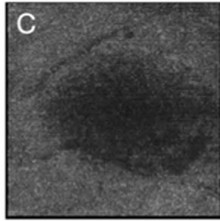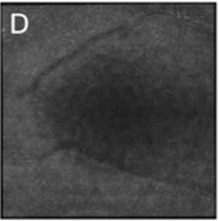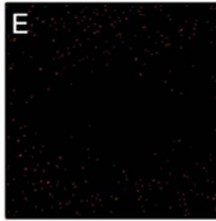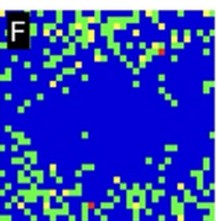

**Fig 1. Basic imaging processing.** A: Single optical coherence tomography (OCT) angiography (OCTA) at the level of the superficial capillary plexus of the left eye of a patient with multiple sclerosis. B: Averaged image of 6 consecutive OCTA after registration. C: Single en-face OCT at the level of the vitreoretinal interface (3 mm slab). D: Averaged en face OCT after registration (using OCTA transform). E: Macrophages like cells (MLC) after removing background and thresholding. F: Density map of MLC (blue: no MLC, green: 1 MLC, yellow: 2 MLC, red: 3 MLC, white: 4 or more MLC).

or unspecified categories. The average time since disease onset was 6.92 years, with a standard deviation of 5.15 years. Regarding disease-modifying treatments, the majority of patients were on ocrelizumab (65%), followed by dimethyl fumarate (19%), ofatumumab (8%) and fingolimod (3%), with other treatments representing smaller proportions.

The EDSS scores in the MS group demonstrated a wide range of neurological disability levels, with a significant proportion of patients presenting minimal or no disability. Nearly half of the patients (45.0%) had an EDSS score of 0.0, indicating no measurable disability, while 16.0% had an EDSS score of 1.0, suggesting mild symptoms without functional impairment. Scores of 1.5 and 2.0 were observed in 10.0% and 6.0% of patients, respectively, reflecting mild disability in specific functional systems. A smaller proportion of patients had moderate disability, with scores of 2.5, 3.0, and 3.5, observed in 4.0%, 4.0%, and 3.0% of patients, respectively. Advanced disability was less common, with scores of 4.0 (4.0%), 4.5 (2.0%), 5.0 (1.0%), and 6.0 or higher (4.0%).

The mean count of T1 lesions in our cohort was 4.1 (SD = 7.4), while the mean T2 lesions was 19.93 (SD = 15.1).

In our cohort, the correlation analysis between the EDSS score and lesion counts in the MS group revealed a weak positive relationship with hypointense T1 lesion count (r = 0.200) and a moderate positive relationship with T2 lesion count (r = 0.365).

The average time for the 9-hole Peg Test was 23.13 seconds (SD = 6.10) for the dominant hand and 25.06 seconds (SD = 6.56) for the non-dominant hand. The average time for the 25-foot walk was 6.56 seconds (SD = 2.13) for trial 1-in and 15.04 seconds (SD = 84.19) for trial 2, with substantial variability observed in the latter.

The analysis of the correlation between EDSS scores and the 9-Hole Peg Test times in the MS group revealed moderate positive relationships for both hands (r = 0.417).

The analysis of the correlation between EDSS scores and 25-foot walk times in the MS group revealed a moderate positive relationship for the 25-foot trial 1 and 2 (r = 0.452).

## Comparison of ophthalmic parameters between the two groups

The average BCVA in the MS group was 0.027 logMAR (SD = 0.072), while in the control group, it was 0.013 logMAR (SD = 0.057). There was no significant difference in BCVA (p = 0.45) between the two groups.

However, MS patients showed significantly lower GCL thickness in the inferior region (p = 0.02) and nasal region (p = 0.006) than controls. In contrast, the superior and temporal areas did not differ significantly (p = 0.12 and p = 0.45, respectively). RNFL measurements showed no statistically significant differences between MS patients and controls across all regions analyzed.

The average MLC parameters in the MS group were a count of 469.93 (SD = 161.24), size of 35.79 (SD = 415.86), and density of 12.86 (SD = 4.72), while in the control group, the average MLC count was 460.68 (SD = 155.54), size was 6.12 (SD = 0.40), and density was 12.29 (SD = 4.92).

The comparison of MLC parameters, including count, size, and density, between MS patients and healthy controls revealed no statistically significant differences. Specifically, the MLC count did not differ significantly between the two groups (p = 0.88), nor did the MLC size (p = 0.48) or MLC density (p = 0.87). **Comparison 1: MS patients with EDSS ≥5 versus healthy controls.** The comparison of MLC parameters between MS patients with an EDSS score of 5 or more and control patients showed a significant difference in MLC size (t = 2.67, p = 0.009), with MS patients with EDSS ≥5 exhibiting larger MLC sizes. No significant differences were observed between these groups for MLC count (t = 1.36, p = 0.176) or MLC density (t = 0.64, p = 0.526).

## Correlation of microglia and MS parameters

The averaged MLC count showed a correlation coefficient of r = 0.115 with the number of T1 lesions in MS patients, while the averaged MLC size had a slightly stronger correlation of r = 0.244. The averaged MLC density also

demonstrated a weak correlation of r = 0.115. On the other hand, MLC size seems to be a parameter of interest in understanding its association with T2 lesion burden in MS patients, as averaged MLC size exhibited a weak positive correlation (r = 0.196, 95% CI: −0.001–0.379, p = 0.051, n = 99; Supplementary S1 Fig), suggesting a trend toward a relationship between larger MLC size and increased T2 lesion count that did not reach statistical significance. Averaged MLC count and density demonstrated weak negative correlations with the number of T2 lesions (r=−0.110 for both), suggesting minimal association.

**Comparison 2: MS patients with EDSS >5 versus MS patients with EDSS ≤5.** The analysis comparing averaged MLC parameters between MS patients with EDSS scores greater than 5 and those with scores of 5 or lower revealed significant differences in MLC count anddensity. Of note, this comparison differs from the one reported above, which compared MS patients with EDSS ≥5 against healthy controls. Patients with EDSS scores over 5 exhibited higher MLC counts and densities, with both parameters showing statistically significant differences (p = 0.020) (Fig 2).

This suggests a potential association between these MLC parameters and greater disability levels in MS. In contrast, no significant difference was observed in MLC size (p = 0.839) between the two groups, indicating that MLC size is unaffected by disability severity as measured by EDSS (Fig 3). Of note, a single data entry outlier with an MLC size of 5843 µm² (compared to a cohort mean of approximately 6–7 µm²) was identified in one eye of a patient with EDSS < 5, which inflated the mean MLC size in that subgroup. A sensitivity analysis excluding this outlier confirmed that the pattern of results remained consistent.

MLC count demonstrated weak negative correlations with GCL measurements, including the inferior (r=−0.147), superior (r=−0.127), nasal (r=−0.177), and temporal regions (r=−0.144). MLC size exhibited a weak positive correlation with the inferior GCL (r = 0.074), while correlations with other GCL regions were negligible.

The analysis examining the correlation between MLC and RNFL parameters of MS patients revealed weak associations across all comparisons. MLC count showed negative correlations with most RNFL measurements, including the superotemporal

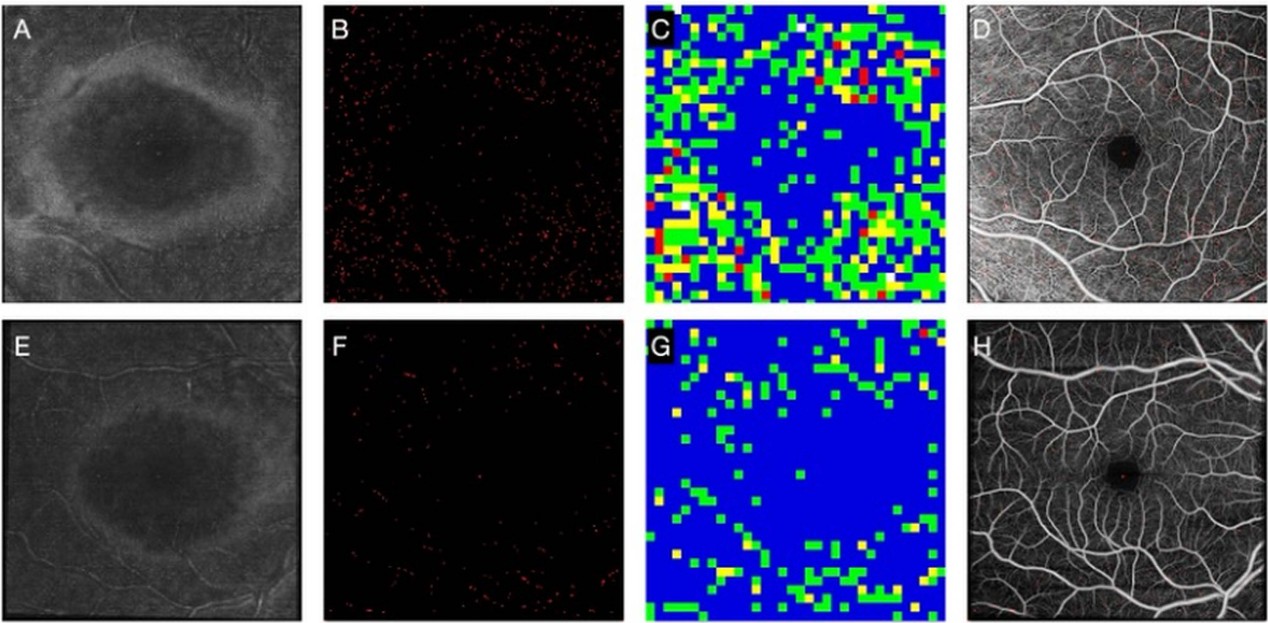

**Fig 2. Example of one patient with multiple sclerosis with a EDSS score of 6.50 showing a higher density of macrophages-like cells (MLC)(A-D), and a control subject (E-H).** A and E: Averaged en-face optical coherence tomography (OCT) at the level of the vitreoretinal interface (3 µm slab). B and F: Thresholded MLC. C and G: Density map of MLC. D and H: Averaged OCT angiography with the added MLC.

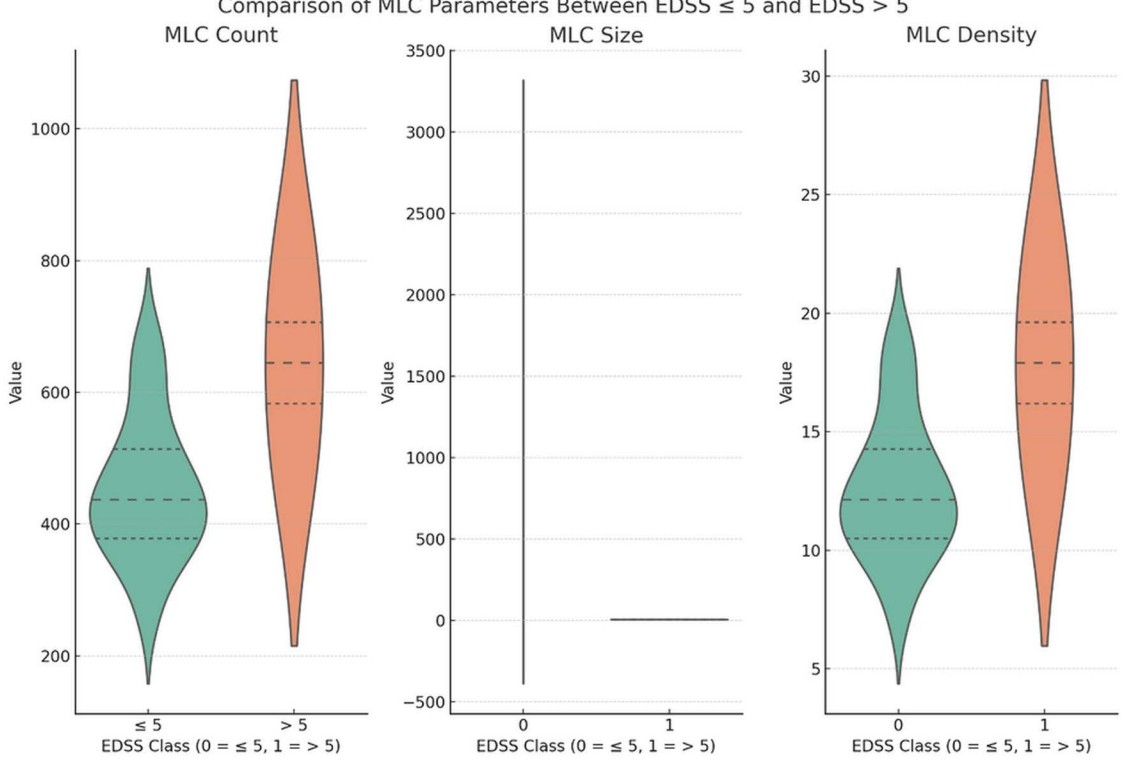

**Fig 3. Enhanced visualization using violin plots to emphasize the differences in MLC parameters between MS patients with EDSS scores ≤ 5 and > 5.** Each plot highlights the distribution of MLC count, size, and density, providing a more precise depiction of significant differences in MLC count and density while showing no substantial variation in MLC size.

(r=−0.139), superonasal (r=−0.129), nasal (r=−0.020), and inferonasal (r=−0.243) regions, while it had a negligible positive correlation with the inferotemporal region (r=0.024). MLC size and density exhibited similarly weak correlations with RNFL parameters, suggesting minimal association between these metrics. These results indicate that MLC parameters do not strongly reflect changes in RNFL thickness, highlighting limited overlap in the biological processes affecting these metrics in MS patients.

The analysis of MLC parameters across different treatment groups in the MS cohort showed no significant differences. Using ANOVA, MLC count (F = 0.878, p = 0.579), MLC size (F = 0.467, p = 0.937), and MLC density (F = 0.693, p = 0.765) were not significantly affected by the type of treatment. These findings suggest that the observed MLC parameters are consistent across treatment modalities in the MS group, indicating that the current disease modifying treatment do not appear to influence these specific metrics.

The analysis comparing MLC parameters based on the onset type in the MS group revealed no statistically significant differences. Using ANOVA, MLC count (F = 0.807, p = 0.702), MLC size (F = 0.475, p = 0.972), and MLC density (F = 0.839, p = 0.665) were found to be consistent across different onset types. These results indicate that the onset type does not appear to influence the measured MLC parameters in the MS cohort.

When comparing MLC parameters with relapsing-remitting MS (RRMS, n = 92) to those with progressive MS (primary or secondary progressive, n = 3) statistical comparison was underpowered; however, visual inspection of boxplots showed no apparent trends distinguishing the two groups.

We further assessed whether disease duration influenced MLC characteristics. Spearman correlation revealed no significant associations between MS duration and MLC count (ρ = 0.086, p = 0.401), MLC size (ρ = 0.145, p = 0.153), or MLC

density (ρ = 0.065, p = 0.523). These findings suggest that in this cohort, neither MS subtype nor disease duration significantly impacted retinal surface microglial parameters.

## Discussion

This study provides exploratory insights into the relationship between clinical disability, imaging parameters, and retinal MLC metrics in patients with MS. Our findings reveal a weak positive trend between retinal MLC size and T2 lesion count (r = 0.196, p = 0.051) that approached but did not reach statistical significance, as well as higher MLC counts and densities among patients with EDSS scores ≥5. The averaged MLC size showed a weak positive correlation with T1 lesion count (r = 0.244), while MLC count and density demonstrated minimal correlations (r = 0.115 each), suggesting a limited association between retinal MLC metrics and the number of T1 hypointense lesions.

Traditionally, T2 lesions reflect areas of demyelination, inflammation, and edema, representing the total burden of disease activity, but lacking specificity for the severity of tissue damage. Some of these lesions, also known as "black holes" appear hypointense on T1-weighted sequences, indicating areas of severe tissue damage, often associated with axonal loss, neurodegeneration and irreversible brain injury.

Our findings align with these established MRI interpretations. In fact, the observed correlation between retinal MLC size and T2 lesion count suggests that microglial activation mirrors overall lesion load and inflammatory processes within the CNS. On the other hand, the weak association between MLC metrics and T1 lesion count may reflect the fact that T1 lesions primarily represent chronic, irreversible tissue damage, whereas MLC metrics may be more sensitive to active or ongoing inflammatory processes rather than established neurodegeneration.

Importantly, the increased MLC counts and densities in patients with higher EDSS scores may reflect a shift toward chronic neuroinflammation and advanced disability.

Taken together, retinal MLC metrics may offer a promising surrogate, non-invasive window into diffuse inflammatory activity (T2 lesions) and, less, into neurodegeneration (T1 black holes), bridging structural imaging with cellular-level insights into disease progression.

Microglia play a pivotal yet complex role in the pathophysiology of MS. Under normal conditions, microglia contribute to CNS homeostasis through immune surveillance and tissue repair. However, in MS, their activity becomes dysregulated, transitioning from a protective role to one that exacerbates neuroinflammation and neuronal damage [27–30]. Early MS lesions are frequently characterized by clusters of reactive microglia, termed microglial nodules, often observed in normal-appearing white matter. These nodules, marked by HLA-DR expression, signify a transition from a surveillant to a reactive, pro-inflammatory phenotype [31].

In contrast to CNS microglia, retinal microglia remain poorly characterized [32]. Retinal microglia constitute approximately 0.2% of retinal cells, primarily located in the Inner Plexiform Layer (IPL) and Outer Plexiform Layer (OPL) [33]. These MLCs include infiltrating macrophages and are increasingly recognized for their role in neuroinflammatory conditions [34,35].

Retinal MLCs can be visualized in vivo using adaptive optics scanning laser ophthalmoscopy (AO-SLO) and OCTA [36]. Techniques such as those described by Castanos et al. and Uji et al. have demonstrated the utility of OCTA in tracking hyperreflective cells over time [37,38]. Liu et al. and other studies have highlighted the significance of MLC density and morphology in neuroinflammatory and vascular conditions, including proliferative diabetic retinopathy, retinal vein occlusion, and uveitis [39]. In uveitis, Pichi et al. reported increased MLC density, size, and number in active posterior uveitis using OCTA [23]. These findings align with those of Zeng et al., who associated elevated MLCs with worsening leakage patterns in Behçet's disease [40]. Such observations suggest that MLCs may be inflammation and disease activity biomarkers, reflecting resident microglial responses and circulating monocyte recruitment.

In our study, patients with more T2 lesions exhibited significantly larger MLC sizes, reflecting hypertrophy consistent with microglial activation in response to inflammation and neurodegeneration. This observation aligns with findings by

Frischer et al., who demonstrated that activated microglia in MS lesions exhibit hypertrophic cell bodies and an amoeboid morphology associated with heightened inflammatory activity and axonal injury [41,42]. The association we observed between MLC size and T2 lesions further underscores microglial activation's role as a CNS lesion burden driver. These findings highlight the potential of retinal MLC metrics as non-invasive biomarkers, reflecting the underlying neurodegenerative processes in MS.

Furthermore, the density of microglia and macrophages increases progressively with disease duration and lesion chronicity, especially in progressive forms of MS. This progressive accumulation, particularly in chronic active lesions with a peripheral rim of activated microglia, is a key factor in the transition from relapsing-remitting to progressive MS [43,44]. Our study's observed MLC counts and densities increase in patients with EDSS ≥5 reflects the connection between microglial activation and disease progression. Specifically, this subgroup exhibited statistically significant differences in MLC counts and densities compared to controls (p = 0.020), while MLC size differences were not significant (p = 0.839). This suggests a complex interplay between microglial proliferation and hypertrophy in advanced disease stages. Similarly, studies like Absinta et al. have shown that chronic active lesions surrounded by activated microglia are strongly linked to disability progression, further supporting the role of microglial activity in MS pathology [45].

All these correlations may point towards future therapeutically targeting microglial activation to mitigate MS pathology. While disease-modifying treatments may effectively modulate clinical disease activity, our study found no significant differences in MLC parameters across treatment groups or onset types. This suggests that current therapies might not significantly impact microglial behavior detectable through imaging techniques. Longitudinal studies employing advanced imaging modalities, such as PET tracers specific for microglial activation, are necessary to explore these relationships further and understand the effects of treatment on microglial dynamics [46–48].

MS has been shown to affect the retina as it extends to the CNS, particularly involving the RNFL and the GCL [49]. Studies by Brandt et al. and Saidha et al. have sparked a discussion regarding primary neurodegeneration in the myelin-deficient retina of MS patients, indicating it occurs independently of optic nerve pathology [50,51]. The neuroaxonal damage caused by MS manifests as thinning of the GCL and the inner plexiform layer, which contains retinal ganglion cell bodies and the peripapillary RNFL composed of unmyelinated axons. This thinning correlates with neuroaxonal loss and physical disability in patients with relapsing-remitting multiple sclerosis [52]. Noninvasive analysis of these retinal layers, particularly GCL, using OCT has proven effective as an MS biomarker [53].

As a highly vulnerable layer to inflammatory and degenerative processes, its thinning indicates neuronal loss and axonal damage [54,55]. Our study demonstrated significant thinning of the GCL, particularly in the inferior and nasal regions, consistent with previous research linking GCL alterations to neurodegeneration in MS. Our findings are consistent with those reported by Bsteh et al., who identified GCL thinning as a strong predictor of disability progression in early relapsing-remitting MS [56]. Additionally, a meta-analysis by Britze et al. showed significantly decreased GCL thickness in MS patients, both with and without prior ON. Most studies (n = 10) reported a strong correlation between GCL thinning and visual function, while several others (n = 6) also associated GCL thinning with expanded disability status scale (EDSS) scores [57]. Lambe et al. (2021) found that a baseline GCL thickness of less than 70 μm was associated with a fourfold increased likelihood of significant worsening in the EDSS [58]. As for the predominantly nasal thinning of the GCL in our MS patients, Shi et al. identified a distinct horseshoe-shaped thinning of the macular ganglion cell and inner plexiform layer in the nasal sector (Zone M), which was strongly associated with visual dysfunction and disability in patients with multiple sclerosis (MS). This thinning pattern suggests that Zone M's location may reflect the papillomacular bundle's preferential involvement, particularly within the parvocellular pathway [59].

Our findings emphasize the potential of retinal imaging as a non-invasive tool for monitoring disease progression in MS. The moderate correlations between functional metrics, including the 9-Hole Peg Test (r = 0.417 for the right hand, r = 0.377 for the left hand) and the 25-foot walk trial 1 (r = 0.452), with EDSS scores in our study, further highlight their relevance in assessing disability. These findings align with prior studies, such as those by Benedict et al. and Motl et al., which

established these tests as reliable indicators of upper and lower limb functionality in MS patients [60,61]. In contrast, the negligible correlation with the 25-foot walk trial 2 (r = 0.034) may reflect external environmental influences or testing conditions, reducing its utility as an intrinsic disability marker.

However, our study is subject to several limitations that warrant consideration.

A key limitation is the quality of the MRI data. Due to unavailability of post-processing software at our center, we were unable to obtain lesion volumes and more advanced quantitative measures, and could only assess lesion count. Moreover, we only analyzed the brain MRI and not the spinal cord MRI of MS patients. This limitation in the MRI data may affect the robustness of our findings [62].

Another limitation involves the sample size of MS patients, which was reduced due to excluding individuals with poor-quality OCT-A scans. These exclusions were necessitated by factors such as nystagmus, diabetic retinopathy, or anterior segment opacities, but they constrain the dataset and may introduce selection bias. Furthermore, as this is a single-center study with a relatively small cohort, the generalizability of the results to larger, more heterogeneous populations is inherently restricted.

The choice of control subjects might represent a bias, as most of them were recruited from the neurology clinic among patients with non-specific complaints and headaches. This may pose a challenge, given the potential neurovascular changes associated with migraines. These changes could act as confounding factors, complicating the interpretation of findings and potentially obscuring differences specific to MS pathology.

Finally, the study's cross-sectional design inherently limits the ability to establish causative links between the observed changes and disease progression. An additional important limitation is that OCT/OCTA imaging cannot definitively identify the cellular origin of the hyperreflective particles detected at the vitreoretinal interface. While the morphological criteria and localization used in our pipeline are consistent with those validated in prior studies using adaptive optics scanning laser ophthalmoscopy (Castanos et al., 2020; Hammer et al., 2023), the detected structures may include resident microglia, infiltrating macrophages, hyalocytes, or other myeloid cells. Direct histological validation through correlative OCT imaging of post-mortem human eyes with whole-mount immunolabeling for microglial/myeloid markers (e.g., Iba-1, CD68) and vascular markers (e.g., CD31/Lectin) would be needed to confirm cell identity. Furthermore, for GCL and RNFL comparisons, eye-level data were used without fully accounting for within-subject correlation between fellow eyes, which may have inflated statistical significance. Additionally, the reliance on lesion counts rather than volumetric MRI measures may have reduced sensitivity in detecting meaningful structure–function relationships, as lesion counts do not capture individual lesion size variability. Future studies should incorporate volumetric MRI assessments, including lesion volume, brain atrophy measures, and spatial lesion mapping. Longitudinal studies with larger, more diverse cohorts and standardized methodologies will be crucial for validating and expanding the findings.

In conclusion, this study highlights the exploratory relationships between clinical disability, retinal imaging metrics, lesion burden, and macrophage-like cell metrics in MS. Retinal MLC metrics, particularly size and density, may warrant further investigation as candidate biomarkers for advanced disease stages, though these findings remain preliminary and require validation in larger cohorts. At the same time, GCL thinning offers a reliable indicator of neurodegeneration. An important caveat is that the cellular identity of the hyperreflective structures detected by OCT at the vitreoretinal interface cannot be confirmed without histological validation; these structures may include resident microglia, infiltrating macrophages, hyalocytes, or other myeloid cells. Correlative post-mortem OCT imaging with whole-mount immunolabeling for microglial and myeloid markers represents a critical next step for biological validation. Future studies should focus on integrating advanced imaging and molecular techniques to validate these findings and explore their implications for disease monitoring and therapeutic strategies.

## Supporting information

**S1 Fig. Scatter plot of averaged macrophage-like cell size versus T2 lesion count in patients with multiple sclerosis (n = 99).** Pearson correlation: r = 0.196, 95% CI: −0.001 to 0.379, p = 0.051.
(TIF)

**S1 Questionnaire. Inclusivity in Global Research questionnaire.**
(DOCX)

## Acknowledgments

The authors thank the Neurology Nursing Department at Cleveland Clinic Abu Dhabi for their continuous support and the Ophthalmic Imaging Team—particularly Hanna Chaudhry, Steven Hay, and Rex Willis—for their technical assistance in obtaining high-quality retinal scans.

## Author contributions

**Conceptualization:** Francesco Pichi, Piergiorgio Neri, Beatrice Benedetti, Ester Carreño.

**Data curation:** Yanny Perez Jimenez, Matteo Belletti, Alia Alsuwaidi, Fatema Alawadhi, Gina Lee, Victoria Mifsud, Beatrice Benedetti, Ester Carreño.

**Formal analysis:** Francesco Pichi, Piergiorgio Neri, Ester Carreño.

**Investigation:** Yanny Perez Jimenez, Gina Lee.

**Methodology:** Francesco Pichi, Yanny Perez Jimenez, Alia Alsuwaidi, Fatema Alawadhi, Piergiorgio Neri, Victoria Mifsud, Anu Jacob, Beatrice Benedetti.

**Project administration:** Gina Lee, Anu Jacob, Beatrice Benedetti.

**Resources:** Yanny Perez Jimenez.

**Software:** Francesco Pichi, Ester Carreño.

**Supervision:** Francesco Pichi, Piergiorgio Neri, Victoria Mifsud, Anu Jacob, Beatrice Benedetti.

**Validation:** Matteo Belletti, Fatema Alawadhi, Ester Carreño.

**Visualization:** Alia Alsuwaidi, Anu Jacob.

**Writing – original draft:** Yanny Perez Jimenez, Matteo Belletti, Alia Alsuwaidi, Fatema Alawadhi, Gina Lee, Piergiorgio Neri, Victoria Mifsud.

**Writing – review & editing:** Francesco Pichi, Piergiorgio Neri, Anu Jacob, Beatrice Benedetti, Ester Carreño.

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
