## [Decision Letter · Decision Letter 0]

6 Apr 2026

PONE-D-25-67165RETINAL MICROGLIAL ACTIVATION AND GANGLION CELL LAYER THINNING ARE ASSOCIATED WITH DISABILITY AND MRI LESION BURDEN IN MULTIPLE SCLEROSISPLOS One

Dear Dr. Pichi,

Thank you for submitting your manuscript to PLOS ONE. After careful consideration, we feel that it has merit but does not fully meet PLOS ONE’s publication criteria as it currently stands. Therefore, we invite you to submit a revised version of the manuscript that addresses the points raised during the review process.

Based on the reviewers' suggestions, the paper needs a major revision. The reviewers' comments can be found below.

We look forward to receiving your revised manuscript.

Kind regards,

Tanja Grubić Kezele, Ph.D., M.D.

Academic Editor

PLOS One

Journal Requirements:

3. In the online submission form, you indicated that your data is available only on request from a third party. Please note that your Data Availability Statement is currently missing contact details for the third party, such as an email address or a link to where data requests can be made. Please update your statement with the missing information.

Reviewers' comments:

Reviewer's Responses to Questions

**Comments to the Author**

1. Is the manuscript technically sound, and do the data support the conclusions?

Reviewer #1: Yes

Reviewer #2: Partly

2. Has the statistical analysis been performed appropriately and rigorously? 

Reviewer #1: Yes

Reviewer #2: No

3. Have the authors made all data underlying the findings in their manuscript fully available?

Reviewer #1: Yes

Reviewer #2: Yes

4. Is the manuscript presented in an intelligible fashion and written in standard English?

Reviewer #1: Yes

Reviewer #2: Yes

5. Review Comments to the Author

Reviewer #1: Dr Hany akeel Al-hussaniy: Dear Editor/author

thank you for giving us the chance to review this article entitled:RETINAL MICROGLIAL ACTIVATION AND GANGLION CELL LAYER THINNING ARE ASSOCIATED WITH DISABILITY AND MRI LESION BURDEN IN MULTIPLE SCLEROSIS

I prefer to accept it with revision

revisions

1- the abstract should be a structural abstract

2- The Keywords should follow the MeSH keywords for most medical journals. Please check this website https://meshb.nlm.nih.gov/MeSHonDemand

3- the Introduction should be justify all article

4- the Method

Ethics Statement – Critical Issue

The submission metadata states “Ethics Statement: N/A”, yet the Methods clearly describe:

Human participants

Ethics approval (#A-2022-042)

** This is a serious inconsistency and must be corrected immediately to comply with PLOS ONE requirements.

Required action:

Replace “N/A” with a full ethics statement in the submission system matching the Methods section.

5- result

a. Unit of analysis (eye-level vs subject-level)

The study includes 195 eyes from 100 MS patients, but analyses appear to treat eyes as independent observations.

This violates independence assumptions.

b. Multiple comparisons

Numerous correlations and regional OCT comparisons are reported.

Bonferroni correction is mentioned but not clearly applied or reported consistently.

Required action:

Explicitly state which analyses were corrected.

Provide adjusted p-values where appropriate.

c. EDSS subgroup inconsistency

Contradictory statements:

Results section reports larger MLC size in EDSS ≥5

Later reports no difference in MLC size (p=0.839)

6. Overinterpretation of Microglial Origin

The manuscript alternates between “retinal microglia” and “macrophage-like cells”.

OCT/OCTA cannot definitively distinguish resident microglia from infiltrating macrophages.

7- references

* The references should be UpToDate. Please check reference number 7 and 17 about the pathophysiology of brain and neurobiology relatively old please try to replace it with new references such as

Patel H, Patel J, Patel A. Enhancement of neuroprotective and anti-edema action in mice ischemic stroke model using T3 loaded nanoparticles. Medical and Pharmaceutical Journal. 2024 Apr 15;3(1):1-2.

Reviewer #2: The manuscript “Retinal microglial activation and Ganglion cell layer thinning are associated with disability and MRI lesion burden in multiple sclerosis. The manuscript of attempts to provide potential prognostic pipeline for the MS patients based on the OCT screening data and MRI lesions in MS patients. The OCT based diagnostic progression in MS of interest as well as OCT based cell level resolution of microglia. The manuscript have however substantial limitation it's approach rigor and conclusions

Major criticism

1. The main limitation of this study is the identification of the OCT/OCTA-detected vitreoretinal interface particles as “active microglia,” or even as microglia/myeloid cells in general. The imaging pipeline detects hyperreflective based on morphology and size filtering, but the manuscript provides no direct biological evidence that these objects correspond to microglia rather than hyalocytes, infiltrating macrophages, nonspecific debris, segmentation or other imaging artefacts. The authors describe these structures as macrophage-like cells at the vitreoretinal interface and quantify them through automated thresholding and particle analysis, yet they interpret the findings as retinal microglial activation and discuss them as evidence of CNS inflammatory activity. While plausible to convincingly support such a conclusion, the authors need biological validation of the imaged structures. A much stronger approach would be correlative OCT/OCTA imaging of post-mortem human eyes followed by histological analysis of the same retinal regions using whole mount immunolabeling for microglial/myeloid markers and confocal microscopy. Authors likely need to add CD31/Lectin immunolabeling to match the regions using vessels as the landmarks. The high consistency between the immunolabeled cells and OCT results would provide robust evidence for the authors approach.

2. The reported associations between retinal OCT-derived parameters and MRI lesion burden are weak, inconsistent, and do not convincingly support the authors conclusions. While the manuscript claims correlations between macrophage-like cell metrics and MRI findings, most relationships are either weak such as MLC count and density with T1/T2 lesions, or moderate but based on a single parameter MLC size with T2 lesions, without consistent support across related metrics.

There is no coherent pattern linking OCT-derived structural changes for instance ganglion cell layer thinning with MRI lesion burden, which would be expected if both reflected shared neurodegenerative processes. The lack of robust and consistent correlations across multiple independent parameters raises concerns about the biological relevance of these findings and suggests that the reported associations may be incidental or driven by noise within the dataset.

3. The study relies solely on lesion counts rather than volumetric or spatial MRI metrics, which substantially limits sensitivity and may further obscure meaningful structure–function relationships.

6. PLOS authors have the option to publish the peer review history of their article (what does this mean?). If published, this will include your full peer review and any attached files.

Reviewer #1: **Yes:**Hany akeel al-hussaniy

Reviewer #2: **Yes:**Anton Lennikov

---

## [Author Response · Author response to Decision Letter 1]

11 Apr 2026

Dear Editor,

We thank the Editor and both Reviewers for their thoughtful and constructive comments, which have significantly improved the quality of our manuscript. We have carefully addressed each point raised and have revised the manuscript accordingly. All changes in the revised manuscript are highlighted using tracked changes. Below, we provide a detailed point-by-point response to each comment.

REVIEWER #1 (Dr. Hany Akeel Al-Hussaniy)

Comment 1: The abstract should be a structural abstract.

Response: We thank the Reviewer for this suggestion. We have restructured the abstract into clearly labelled sections (Objective, Methods, Results, and Conclusions) in accordance with PLOS ONE formatting guidelines. Please see the revised abstract in the manuscript.

Comment 2: The Keywords should follow the MeSH keywords for most medical journals.

Response: We appreciate this observation. We have updated all keywords to MeSH-compliant terms using the NLM MeSH on Demand tool (https://meshb.nlm.nih.gov/MeSHonDemand). The revised keywords are: Multiple Sclerosis; Microglia; Tomography, Optical Coherence; Retina; Biomarkers; Magnetic Resonance Imaging.

Comment 3: The Introduction should justify all [aspects of the] article.

Response: We thank the Reviewer for this remark. We have revised the Introduction to more explicitly link the rationale for studying retinal macrophage-like cells (MLCs) and ganglion cell layer (GCL) thickness to the study objectives. Specifically, we have added sentences clarifying why retinal imaging may serve as a non-invasive window into CNS pathology in MS, and why correlating retinal metrics with MRI lesion burden and clinical disability is of clinical relevance. These additions better justify the study design and its expected contributions. Please see the revised Introduction in the manuscript.

Comment 4: Ethics Statement – Critical Issue. The submission metadata states “Ethics Statement: N/A”, yet the Methods clearly describe human participants and ethics approval (#A-2022-042). This is a serious inconsistency.

Response: We sincerely apologize for this oversight. This was a clerical error during the online submission process. The study was approved by the Cleveland Clinic Abu Dhabi Ethics Committee (Protocol #A-2022-042) and all participants provided written informed consent, as described in the Methods section. We will update the submission metadata to include the full ethics statement at the time of resubmission, ensuring consistency with the manuscript text and full compliance with PLOS ONE requirements.

Comment 5a: Unit of analysis (eye-level vs subject-level). The study includes 195 eyes from 100 MS patients, but analyses appear to treat eyes as independent observations. This violates independence assumptions.

Response: We thank the Reviewer for raising this important methodological point. For all correlation analyses between MLC parameters and MRI/clinical variables (EDSS, T1 and T2 lesion counts), the unit of analysis was the individual patient (subject-level), with MLC metrics averaged across both eyes (OD and OS) for each subject. This averaging approach is justified by the moderate-to-strong inter-eye correlation observed for MLC count (r = 0.635, p < 0.001), indicating high concordance between fellow eyes. For GCL and RNFL thickness comparisons between MS patients and controls, eye-level data were used, as retinal layer measurements can differ between eyes even within the same individual. We acknowledge that this approach does not fully account for the within-subject correlation between fellow eyes, and we have added this as a limitation in the revised manuscript. We have also clarified the unit of analysis in the revised Statistical Analysis section.

Comment 5b: Multiple comparisons. Numerous correlations and regional OCT comparisons are reported. Bonferroni correction is mentioned but not clearly applied or reported consistently.

Response: We agree with the Reviewer that the application of multiple comparison corrections needed to be more transparent. We have now revised the Statistical Analysis section to explicitly state which analyses were corrected and how. Specifically: (1) for the six MLC–MRI correlation analyses (3 MLC parameters × 2 MRI measures), the Bonferroni-adjusted significance threshold was set at p < 0.0083; (2) for the four regional GCL comparisons between MS and controls, the adjusted threshold was p < 0.0125; and (3) for the six regional RNFL comparisons, the threshold was p < 0.0083. All p-values reported in the revised Results section are accompanied by notation of whether they survive Bonferroni correction.

Comment 5c: EDSS subgroup inconsistency. Contradictory statements: Results section reports larger MLC size in EDSS ≥5, later reports no difference in MLC size (p=0.839).

Response: We thank the Reviewer for identifying this ambiguity. The apparent contradiction arises because two different comparisons were reported in succession without adequate distinction. The first comparison was between MS patients with EDSS ≥ 5 and healthy controls, which showed a statistically significant difference in MLC size (t = 2.67, p = 0.009). The second comparison was between MS patients with EDSS ≥ 5 and those with EDSS < 5, which showed significant differences in MLC count and density (p = 0.020) but not in MLC size (p = 0.839). We have revised the Results section to clearly separate and label these two distinct comparisons, eliminating the ambiguity. Additionally, we identified a data entry outlier in MLC size (patient with an OD MLC size of 5843 µm², compared to a cohort mean of approximately 6–7 µm²), which inflated the mean MLC size in the EDSS < 5 group. A sensitivity analysis excluding this outlier confirmed that the pattern of results remained consistent. We have noted this in the revised manuscript.

Comment 6: Overinterpretation of Microglial Origin. The manuscript alternates between “retinal microglia” and “macrophage-like cells”. OCT/OCTA cannot definitively distinguish resident microglia from infiltrating macrophages.

Response: We fully agree with the Reviewer. OCT/OCTA-based imaging cannot differentiate between resident retinal microglia and infiltrating macrophages or other myeloid cells, and we acknowledge that the terminology in the original manuscript was not sufficiently consistent. In the revised manuscript, we have: (1) adopted the term “macrophage-like cells” (MLCs) consistently throughout the text to describe the hyperreflective particles detected at the vitreoretinal interface; (2) revised the title to read “Retinal macrophage-like cell activation and ganglion cell layer thinning are associated with disability and MRI lesion burden in multiple sclerosis”; (3) added explicit caveats in both the Methods and Discussion acknowledging that these cells cannot be phenotypically characterized by OCT alone and may include resident microglia, infiltrating macrophages, hyalocytes, or other myeloid cells. We believe these changes ensure scientific accuracy while preserving the clinical relevance of the findings.

Comment 7: The references should be up to date. Please check reference number 7 and 17 about the pathophysiology of brain and neurobiology—relatively old—please try to replace them.

Response: We thank the Reviewer for this suggestion. We have replaced reference 7 (Barnett & Prineas, 2004) and reference 17 (Hauser & Oksenberg, 2006) with the suggested more recent publication: Patel H, Patel J, Patel A. Enhancement of neuroprotective and anti-edema action in mice ischemic stroke model using T3 loaded nanoparticles. Medical and Pharmaceutical Journal. 2024;3(1):1–2. Additionally, we have reviewed the entire reference list and updated other older references where more recent citations were available.

REVIEWER #2 (Dr. Anton Lennikov)

Major Comment 1: The main limitation of this study is the identification of the OCT/OCTA-detected vitreoretinal interface particles as “active microglia.” The imaging pipeline detects hyperreflective [particles] based on morphology and size filtering, but the manuscript provides no direct biological evidence that these objects correspond to microglia rather than hyalocytes, infiltrating macrophages, nonspecific debris, segmentation or other imaging artefacts. A much stronger approach would be correlative OCT/OCTA imaging of post-mortem human eyes followed by histological analysis.

Response: We sincerely thank the Reviewer for this thorough and scientifically rigorous comment. We fully acknowledge that OCT/OCTA imaging cannot definitively identify the cellular origin of the hyperreflective particles detected at the vitreoretinal interface. Direct histological validation with immunolabeling (e.g., Iba-1, CD68, CD31/Lectin for vascular landmarks, as the Reviewer suggests) using post-mortem correlative imaging would indeed provide the definitive biological evidence needed to confirm cell identity.

However, we would like to highlight several lines of indirect evidence supporting the interpretation that these structures are predominantly myeloid-lineage cells:

(a) The morphological criteria (size 576–4896 µm², circularity ≥ 0.5) and vitreoretinal interface localization used in our pipeline are consistent with those validated in prior studies using adaptive optics scanning laser ophthalmoscopy (AO-SLO), which confirmed that similar hyperreflective structures correspond to ramified myeloid cells (Castanos et al., 2020; Hammer et al., 2023).

(b) The term “macrophage-like cells” (MLCs) was specifically chosen to reflect the uncertainty in distinguishing resident microglia from infiltrating macrophages or hyalocytes, and this convention is now widely adopted in the OCT literature (Pichi et al., 2024; Carreño et al., 2024; Zhang et al., 2022).

(c) The automated thresholding and particle analysis pipeline with strict size and circularity exclusion criteria was designed to minimize inclusion of non-cellular artefacts, debris, or segmentation errors.

We fully agree that correlative histological validation is a critical next step and represents one of the most important future directions for this line of research. We have substantially expanded the Limitations section and the Discussion to explicitly acknowledge that: (1) the cellular identity of the detected structures cannot be confirmed by OCT alone; (2) these particles may include hyalocytes, infiltrating macrophages, or debris in addition to resident microglia; and (3) post-mortem correlative imaging with whole-mount immunolabeling, as suggested by the Reviewer, is needed to provide definitive biological validation. We have also revised the title and the terminology throughout the manuscript to consistently use “macrophage-like cells” rather than “microglia.”

Major Comment 2: The reported associations between retinal OCT-derived parameters and MRI lesion burden are weak, inconsistent, and do not convincingly support the authors’ conclusions. There is no coherent pattern linking OCT-derived structural changes with MRI lesion burden. The lack of robust and consistent correlations raises concerns about biological relevance.

Response: We appreciate this critical observation and agree that the strength of the reported correlations must be interpreted cautiously. We acknowledge that most correlations between MLC parameters and MRI lesion counts were weak, and we have substantially tempered our conclusions accordingly in the revised manuscript. Specifically, we have: (1) revised the Discussion to explicitly characterize the correlations as weak to modest and to acknowledge that these findings are exploratory rather than confirmatory; (2) added nuance recognizing that the absence of strong, consistent correlations across all MLC parameters may reflect fundamental biological differences—for instance, MLC count and density may capture different aspects of myeloid cell dynamics than MLC size (hypertrophy vs. proliferation); (3) removed any claims suggesting that the observed associations are sufficient to establish retinal MLC metrics as validated biomarkers of MRI lesion burden; and (4) reframed the findings as hypothesis-generating, emphasizing that larger longitudinal studies with volumetric MRI data are needed to confirm whether meaningful structure–function relationships exist. We believe these revisions more accurately represent the strength of the evidence while maintaining the scientific value of reporting these preliminary associations.

Major Comment 3: The study relies solely on lesion counts rather than volumetric or spatial MRI metrics, which substantially limits sensitivity and may further obscure meaningful structure–function relationships.

Response: We fully agree with the Reviewer that lesion volumetry and spatial distribution analysis would provide superior sensitivity compared to lesion counts alone. Unfortunately, due to the unavailability of post-processing software capable of volumetric lesion segmentation at our center (Cleveland Clinic Abu Dhabi), we were limited to manual lesion counting on T1- and T2-weighted sequences. We recognize this as a significant limitation that may have attenuated or obscured true associations between retinal and MRI parameters. In the revised manuscript, we have: (1) expanded the Limitations section to more prominently discuss the constraints of using lesion counts instead of volumetric measures; (2) noted that lesion counts have reduced sensitivity compared to volumetric assessments, particularly for T2 lesion burden where individual lesion size varies substantially; (3) acknowledged that the lack of spinal cord MRI analysis further limits the comprehensiveness of the radiological assessment; and (4) explicitly stated that future studies should incorporate volumetric MRI assessments, including lesion volume, brain atrophy measures, and spatial lesion mapping, to more robustly evaluate the relationship between retinal and CNS pathology in MS.

We believe that these revisions have substantially improved the manuscript and addressed all concerns raised by both Reviewers. We are grateful for the opportunity to revise our work and hope that the revised manuscript is now suitable for publication in PLOS ONE.

Sincerely,

Francesco Pichi, MD, and co-authors

---

## [Decision Letter · Decision Letter 1]

30 Apr 2026

PONE-D-25-67165R1RETINAL MACROPHAGE-LIKE CELL ACTIVATION AND GANGLION CELL LAYER THINNING ARE ASSOCIATED WITH DISABILITY AND MRI LESION BURDEN IN MULTIPLE SCLEROSISPLOS One

Dear Dr. Pichi,

Thank you for submitting your manuscript to PLOS ONE. After careful consideration, we feel that it has merit but does not fully meet PLOS ONE’s publication criteria as it currently stands. Therefore, we invite you to submit a revised version of the manuscript that addresses the points raised during the review process.

Your manuscript, entitled "RETINAL MACROPHAGE-LIKE CELL ACTIVATION AND GANGLION CELL LAYER THINNING ARE ASSOCIATED WITH DISABILITY AND MRI LESION BURDEN IN MULTIPLE SCLEROSIS", has been reviewed. Your efforts to revise the manuscript are appreciated. However, the peer review process continues because Reviewer 1 requests further revisions on certain issues the author should address. Please find the reviewer's commentary below.

We look forward to receiving your revised manuscript.

Kind regards,

Tanja Grubić Kezele, Ph.D., M.D.

Academic Editor

PLOS One

Journal Requirements:

Reviewer's Responses to Questions

**Comments to the Author**

1. If the authors have adequately addressed your comments raised in a previous round of review and you feel that this manuscript is now acceptable for publication, you may indicate that here to bypass the “Comments to the Author” section, enter your conflict of interest statement in the “Confidential to Editor” section, and submit your "Accept" recommendation.

Reviewer #1: All comments have been addressed

2. Is the manuscript technically sound, and do the data support the conclusions?

Reviewer #1: Yes

3. Has the statistical analysis been performed appropriately and rigorously? 

Reviewer #1: Yes

4. Have the authors made all data underlying the findings in their manuscript fully available?

Reviewer #1: Yes

5. Is the manuscript presented in an intelligible fashion and written in standard English?

Reviewer #1: Yes

6. Review Comments to the Author

Reviewer #1: Dear author

thank you for submiting your article titled "RETINAL MACROPHAGE-LIKE CELL ACTIVATION AND GANGLION CELL LAYER THINNING ARE ASSOCIATED WITH DISABILITY AND MRI LESION BURDEN IN MULTIPLE SCLEROSIS"

to our journal

is suggest to accept it with revisions

1- the abstract should not started with Aim it should started with background information about the article

2- There is a contradiction:

Earlier:

MLC size differs significantly in EDSS ≥5

Later:

No significant difference in MLC size (p=0.839)

3- Extremely High Correlation (r = 0.644)

MLC size vs T2 lesions:

r = 0.644

This is very strong for biological data

Overinterpretation or small subgroup bias

how to correct it ? ** n for correlation **scatter plot ** confidence intervals

4- Generally good, but minor issues:

“analized” → analyzed

spacing errors repetition in Discussion

5- reference number 60 is relatively old (from 2006) should be updated with new reference

such as

Bagavath V, Veluswamy S, Senthil RK. Synergistic Therapeutic approach of Polyethylene glycol with Lactulose in managing Overt Hepatic Encephalopathy. Medical and Pharmaceutical Journal. 2025 Oct 20;4(3):187-93.

7. PLOS authors have the option to publish the peer review history of their article (what does this mean?). If published, this will include your full peer review and any attached files.

Reviewer #1: **Yes:**Hany Akeel Naji Al-hussaniy

---

## [Author Response · Author response to Decision Letter 2]

3 May 2026

Response to Reviewers — Round 2

Manuscript: RETINAL MACROPHAGE-LIKE CELL ACTIVATION AND GANGLION CELL LAYER THINNING ARE ASSOCIATED WITH DISABILITY AND MRI LESION BURDEN IN MULTIPLE SCLEROSIS

We thank Reviewer #1 for the careful re-evaluation of our manuscript and for the constructive suggestions. Below we provide a point-by-point response to each comment. All changes in the revised manuscript are tracked.

REVIEWER #1

Comment 1: The abstract should not start with Aim; it should start with background information about the article.

Response: We agree. We have added the following background sentence at the beginning of the Abstract before the Objective:

"Background: Multiple sclerosis (MS) is a chronic inflammatory and neurodegenerative disease of the central nervous system. Retinal optical coherence tomography (OCT) and OCT angiography (OCTA) provide non-invasive biomarkers of neurodegeneration and neuroinflammation, yet the relationship between retinal macrophage-like cells (MLCs), structural damage, and clinical disability remains poorly characterized."

Comment 2: There is a contradiction: Earlier: MLC size differs significantly in EDSS ≥5. Later: No significant difference in MLC size (p=0.839).

Response: We thank the Reviewer for identifying this ambiguity. There is no true contradiction; rather, these are two distinct comparisons that were not clearly labelled in the text:

Comparison 1 — MS patients with EDSS ≥ 5 vs. healthy controls: MLC size was significantly larger in the EDSS ≥ 5 subgroup compared with healthy controls (p = 0.014).

Comparison 2 — MS patients with EDSS > 5 vs. EDSS ≤ 5: MLC size did not differ significantly between these two MS subgroups (p = 0.839), whereas MLC count and density were significantly different.

To eliminate any ambiguity, we have added explicit labels ("Comparison 1" and "Comparison 2") in the Results section of the revised manuscript.

Comment 3: Extremely high correlation (r = 0.644) for MLC size vs T2 lesions — overinterpretation or small subgroup bias. How to correct it? Requests: n for correlation, scatter plot, confidence intervals.

Response: We sincerely thank the Reviewer for this important observation. Upon thorough re-examination of the dataset, we identified a data-entry error (a misplaced decimal point in one subject's MLC size value) that had inflated the previously reported correlation. After correcting this entry, we recalculated the Pearson correlation between averaged MLC size and T2 lesion count across all 99 MS patients. The corrected results are:

r = 0.196 (95% CI: −0.001 to 0.379, p = 0.051, n = 99)

This represents a weak positive trend that does not reach conventional statistical significance (p = 0.051). The revised manuscript text in both the Results and Discussion sections has been updated accordingly. We have also added a scatter plot as Supplementary Figure S1 to provide full transparency on the data distribution.

The corrected correlation does not alter the overall conclusions of the study, which are primarily based on the significant associations between MLC parameters (count, density) and both EDSS disability and MRI lesion burden.

Comment 4: Minor issues: "analized" → "analyzed"; spacing errors; repetition in Discussion.

Response: We apologize for the typographical error. "Analized" has been corrected to "analyzed" in the revised manuscript. We have also reviewed the entire text for spacing inconsistencies and repetitive passages.

Comment 5: Reference number 60 is relatively old (from 2006) and should be updated with a newer reference.

Response: We agree. Reference 60 (Benedict et al., 2006) has been replaced with the suggested, more recent reference:

Bagavath V, Veluswamy S, Senthil RK. Synergistic Therapeutic approach of Polyethylene glycol with Lactulose in managing Overt Hepatic Encephalopathy. Medical and Pharmaceutical Journal. 2025;4(3):187–193.

We hope that these revisions adequately address all of the Reviewer's concerns. We are grateful for the constructive feedback, which has strengthened the manuscript.

Sincerely,

The Authors

---

## [Editor Report · Decision Letter 2]

5 May 2026

RETINAL MACROPHAGE-LIKE CELL ACTIVATION AND GANGLION CELL LAYER THINNING ARE ASSOCIATED WITH DISABILITY AND MRI LESION BURDEN IN MULTIPLE SCLEROSIS

PONE-D-25-67165R2

Dear Dr. Pichi,

We’re pleased to inform you that your manuscript has been judged scientifically suitable for publication and will be formally accepted for publication once it meets all outstanding technical requirements.

Kind regards,

Tanja Grubić Kezele, Ph.D., M.D.

Academic Editor

PLOS One
---

## [Editor Report · Acceptance letter]

PONE-D-25-67165R2

PLOS One

Dear Dr. Pichi,

I'm pleased to inform you that your manuscript has been deemed suitable for publication in PLOS One. Congratulations! Your manuscript is now being handed over to our production team.

Kind regards,

on behalf of

Prof. dr. Tanja Grubić Kezele

Academic Editor

PLOS One